# Women empowerment and sexually transmitted infections: Evidence from Bangladesh demographic and health survey 2014

Md Abdullah Al Jubayer Biswas[1], Mohammad Abdullah Kafi[2], Muhammad Manwar Morshed Hemel[1], Mondar Maruf Moin Ahmed[3], Sharful Islam Khan[1]*

1 Program for HIV and AIDS, Infectious Diseases Division, icddr,b, Dhaka, Bangladesh, 2 Program for Emerging Infection, Infectious Diseases Division, icddr,b, Dhaka, Bangladesh, 3 Maternal and Child Nutrition, Nutrition and Clinical Services Division, icddr,b, Dhaka, Bangladesh

☯ These authors contributed equally to this work.

* sharful@icddrb.org

## Abstract

### Background

Sexually transmitted infections (STIs) among women have led to substantial public health and economic burdens in several low-middle-income countries. However, there is a paucity of scientific knowledge about the relationship between empowerment and symptoms of STIs among married Bangladeshi women. This article aimed to examine the association between women empowerment and symptoms of STIs among currently married Bangladeshi women of reproductive age.

### Materials and methods

We extracted data from the Bangladesh Demographic and Health Survey (BDHS), conducted from June 28, 2014, to November 9, 2014. We utilised cross-tabulation, the conceptual framework and multivariable multilevel mixed-effect logistics regression to explore the association between women's empowerment indicators and women's self-reported symptoms of genital sore and abnormal genital discharge. All of the analysis was adjusted using cluster weight.

### Results

We found that among 16,858 currently married women, 5.59% and 10.84% experienced genital sores and abnormal genital discharge during the past 12 months, respectively. Women who depended on husbands to make decisions regarding their health care (AOR = 0.75, 95% CI = 0.67–0.84), significant household purchases (AOR = 0.79, 95% CI = 0.71–0.88), and visiting family or relatives (AOR = 0.72, 95% CI = 0.64–0.80) were less likely to report signs of abnormal genital discharge. Women who could make joint healthcare

**Data Availability Statement:** Data is publicly available on the website of The DHS Program (https://dhsprogram.com/data/available-datasets.

cfm), which can be downloaded based on the request.

**Funding:** The authors received no specific funding for this work.

**Competing interests:** The authors have declared that no competing interests exist

decisions with their husbands were also less likely to report genital sores (AOR = 0.78, 95% CI = 0.67–0.90).

## Conclusion

Genital sores and abnormal genital discharge were prevalent across all parameters of women empowerment among currently married women in Bangladesh. Our estimates show that the husband plays a significant role in decision-making about sexual and reproductive health. Efforts need to be invested in establishing culturally relevant gender policies which facilitate the involvement of women in joint decision-making.

## Introduction

Sexually transmitted infections (STIs) pose substantial public health threats and burdens, particularly among women of reproductive age and their children. According to global estimates, over 900,000 pregnant women are have been infected with *syphilis* [1]. As of 2016, *syphilis* gave rise to approximately 350,000 adverse birth outcomes, including stillbirth [1]. STIs such as *syphilis*, *Chlamydia trachomatis*, *Neisseria gonorrhoeae*, and *Trichomonas vaginalis* have triggered several symptoms, including genital sores and abnormal genital discharge. These STIs are linked to numerous health complications, including the risk of HIV transmission [2–5]. A systematic review based on studies from 30 low- and middle-income countries (LMIC) indicate that STIs among women are quite common across the different geographical settings even if the range of prevalence varies. For example, *Neisseria gonorrhoeae* was found to be 1.2%-4.6%, *syphilis* ranged from 1.1%-6.5%, *Chlamydia trachomatis* was approximately 0.8%-11.2%, and *Trichomonas vaginalis* ranged from 3.9%-24.6% [6]. Moreover, a cohort study among South African women depicted that the presence of genital discharge, genital sores or epithelial disruptions were strongly associated with HIV seroconversion [5]. In LMIC, the substantial STIs burden can be attributed to inadequate healthcare access, delayed or inadequate detection, and economic inequality of women [7, 8]. Moreover, another literature review based in 13 countries showed that Muslim women generally had poor knowledge about STIs signs and symptoms, prevention, diagnosis, and management. The review also revealed misconceptions that fueled blame and judgmental attitudes towards women who were infected [9].

In Bangladesh, a low number of STIs cases are reported due to insufficient knowledge, the social stigma attached to STIs, and a scarcity of affordable sexual healthcare options [10]. However, the 2014 Bangladesh Demographic and Health Survey (BDHS) reported that 10.8% and 5.8% of ever-married (i.e., currently married, divorced, widowed, and separated) women aged 15–49 years experienced genital discharge and genital sores, respectively [11]. Furthermore, literature showed that women were reluctant to disclose sexual health issues and seek healthcare due to the societal stigma associated with sexual health [10]. These social taboos even dissuaded them from confiding about their symptoms of STIs with their intimate partners. These circumstances led to their health issues remaining overlooked, thus gradually becoming chronic and complicated [12].

Women empowerment, which has a multifaceted and nuanced association with health, is crucial for safeguarding their health and their families' welfare [13]. In particular, recent literature demonstrated the relationship between various contexts and dynamics of women empowerment and their health [13–16]. Moreover, several studies indicated that gendered power dynamics in intimate relationships had deterred women from making sexual and reproductive

health (SRH) decisions [17]. Many women living with HIV and STIs cannot exercise their rights to seek SRH-related healthcare due to gender inequality, both within the society and their intimate partnerships [17]. Likewise, evidence from LMIC shows that women empowerment is associated with healthcare-seeking behaviours, the use of contraceptives, household decision-making, employment status, and freedom of movement [18]. In Bangladesh, a similar association was reflected between women's autonomy and healthcare service uptake [13, 19]. As most women in Bangladesh are submissive to their partners' requests, this influences their sexual and reproductive health [20]. Evidence from previous studies in Bangladesh had also indicated that participation in income-generating activities positively affected women's ability to make household decisions such as major purchases, healthcare for themselves and their family members, and engagement in recreational activities [21]. Although they could not assume autonomy over their income, these initiatives allowed them to alleviate previous power imbalances, thus ultimately protecting them from intimate partner violence (IPV) [22]. Thus, self-care empowerment is one of the prerequisites for preventing unsafe sexual behaviours, which could lead to STIs [23]. This type of empowerment could enable women to refer their married partners for STIs care, whereas women who experience economic vulnerability and limited empowerment struggle to guide their partners [24].

Bangladesh has made promising progress in alleviating gender disparity, ranking 68[th] out of 156 countries as per the World Economic Forum's annual gender gap report of 2021 [25]. Women constitute half of the population, and their participation in the workforce has grown exponentially [26]. It would be challenging to accomplish the Sustainable Development Goal (SDG) that pertains to gender equality if the health and wellbeing of a substantial number of women remain overlooked [27]. Therefore, it is integral to ensure autonomy and self-empowerment to effectively negotiate and exercise their sexual and reproductive health rights to their intimate partners. However, in Bangladesh, the relationship between women empowerment and symptoms of STIs have not been thoroughly investigated or evaluated during their reproductive age. Moreover, there is a paucity of primary and secondary evidence in Bangladesh that illuminates the connection between women empowerment and symptoms of STIs.

Therefore, this study aimed to analyse nationally representative population-based data and present findings of the association between women empowerment and self-reported genital discharge and genital sores. The findings from this exercise can assist policymakers in developing gender-sensitive and empowering STIs preventive policies and broad healthcare system policies which could address social health determinants and, thus, ultimately prevent STIs.

## Materials and methods

### Study design and sampling technique

We used data from the BDHS 2014, a nationally representative cross-sectional survey. Since the most recent publicly accessible BDHS 2017 data does not contain STIs and HIV-related data, BDHS 2014 served as the final dataset for our research objective. The survey was conducted between June and November 2014 by the ICF International (USA), National Institute of Population Research and Training (NIPORT) and Mitra and Associates [11]. The sampling method was a two-stage stratified sampling strategy of the households. During the first stage, 600 enumeration areas (EAs), consisting of 207 urban and 393 rural EAs, were selected using the probability-proportionate-to-size approach. An EA was defined as a village, small village or part of a large village. Secondly, a systematic sample of 30 households was selected from each EA to provide statistically reliable estimates of key demographic and health variables [11]. The sampling procedure has been detailed in the BDHS 2014 report [11]. From 18,000 sampled residential homes, 17,300 residential households were surveyed, and 17,886 ever-married

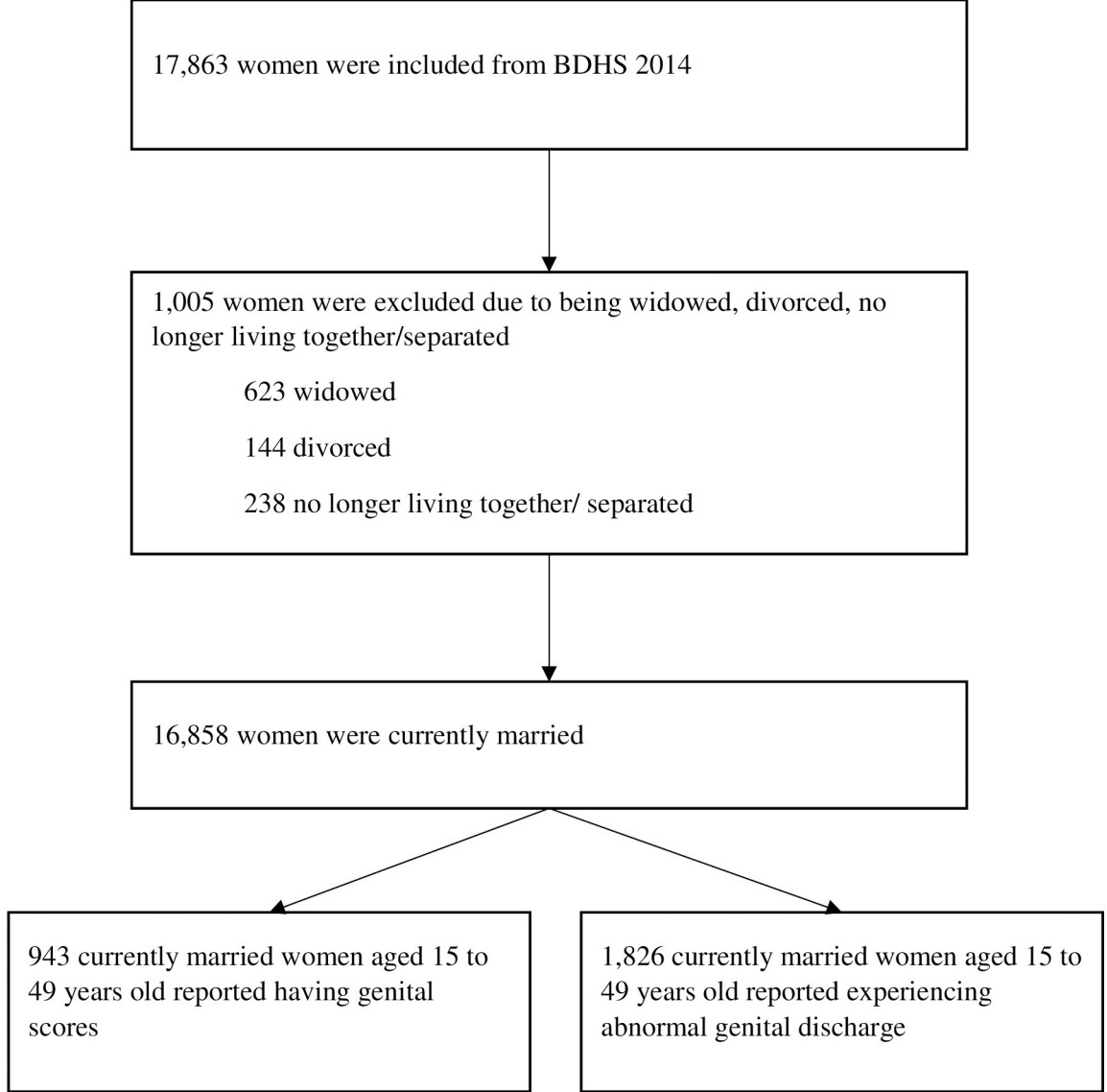

**Fig 1. A flow chart of study population selection from Bangladesh demographic health survey (BDHS) 2014.**

women were interviewed. Women were interviewed using a questionnaire to elicit information about their background characteristics, information about their partners and HIV/AIDS-related information. Informed consent was obtained from the respondents before conducting the interview. The questionnaire was adapted, pretested and validated as per the context of Bangladesh Since its inception, the complete questionnaire and data collection procedure were rendered publicly accessible [11]. This study restricted the analysis to 16,858 currently married women who had experienced abnormal genital discharge or genital sores within the past 12 months preceding data collection (Fig 1).

## Outcome variables

This study examined two different outcome variables (1) genital sores; and (2) abnormal genital discharge among currently married women aged 15–49 years. Table 1 includes a detailed

**Table 1. Explanation of outcome variables.**

| Variables | Explanation |
| --- | --- |
| Abnormal genital discharge | Women aged 15–49 years had abnormal (or bad-smelling) genital discharge in the past 12 months. It was re-coded and categorised as: Yes (Yes), No (No and Don't know) |
| Genital sore | Women aged 15–49 years had a genital sore or ulcer in the past 12 months. It was grouped as Yes (Yes), No (No and Don't know) |

explanation of the two outcome variables. According to the BDHS-2014 survey, abnormal vaginal discharge or presence of genital sore was considered as self-reported STIs or symptoms of STIs [11].

## Explanatory variables

All of the independent variables are illustrated in Table 2. They were chosen based on the existing literature [11, 13, 28–31]. The household wealth index was evaluated by BDHS 2014 using scores derived from the principal component analysis of various household amenities, possessions and assets [11]. In our analysis, women's socio-demographic variables were expressed as age categories, women's education, husband's education, residence, division and wealth index. The indicators of women empowerment included employment status, freedom of movement, control over their earnings, participation in decision-making in the household and acceptance of physical battery of wives (wife-beating) (Table 2).

## Statistical analysis

We analysed all socio-demographic characteristics and women empowerment indicators using descriptive statistics (i.e., frequency, percentage, cross-tabulation, mean and standard deviation) to describe attributes of currently married women and understand the prevalence of the two STIs symptoms. Sampling weight was adopted from the BDHS database and used to conduct all analyses. Based on empirical knowledge from previous research and the authors' scientific expertise, a computational framework approach was used to examine the causal pathway between outcome variables and explanatory variables (Fig 2) [13, 30–35]. In the conceptual framework, women empowerment indicators and women's socio-demographic characteristics were both directly and indirectly linked to genital discharge and genital sores where the single direct arrows depict direct links. Women's demographic characteristics and empowerment indicators indirectly affect the outcomes through safe sex knowledge, practice and health-seeking behaviour. Due to inadequate data, only direct links were considered for model building purposes. For example, as confounders such as age group, women education, husband's education, wealth index, division, exposure to media, place of residence were perceived to affect the imagined relationship between women empowerment indicators and outcomes of interest, they were included in the final multivariable model to control these potential confounding effects.

We examined the relationship between women empowerment indicators and outcome variables using multilevel mixed-effect logistic regression. To demonstrate the clustering of primary sampling units (PSU), we initially ran an empty model/null model with no explanatory variables to investigate the variance of the outcomes. Then, explanatory variables were applied concurrently. The multilevel mixed-effects model consisted of the two effects. The fixed effect equation depicted the relationship between explanatory and outcome variables wherein the random effect was determined using variance and inter-cluster correlation (ICC). The likelihood ratio (LR) test was performed to check model adequacy.

**Table 2. Explanation of explanatory variables.**

| Variables | Explanation |
|---|---|
| Age group (in years) | Self-reported age of women at the time of survey categorised into 15–19 Years; 20–24 Years; 25–29 Years; 30–34 Years; 35–39 Years; 40–44 Years; 45–49 Years |
| Women education | The highest level of education attained by women classified in terms of No education; Primary; Secondary; and Higher |
| Husband's education | The highest level of education attained by husband classified in terms of No Education; Primary; Secondary; Higher |
| Wealth index | The composite index of household goods, services and assets. It was derived using the principal component analysis to produce a standard factor score, divided into five equal parts and grouped as Poorest (Lowest), Poorer (Second), Middle, Richer (Fourth), Richest (Highest) |
| Place of residence | Types of the place of residence: Urban; and Rural |
| Division | The division where the respondent resided at the time of the survey: Barisal; Chittagong; Dhaka; Khulna; Rajshahi; Rangpur; Sylhet |
| Exposure to media | This composite variable derived from three different variables including 1) reading newspaper 2) watching television 3) listening to the radio at least once a week and grouped as: No exposure; Exposure to 1–2 media; Exposure to all three media |
| **Women empowerment indicators** | |
| Employment status | Employment status of the resident at any given time in the past 12 months: Yes (currently working, worked in the past year, presently working or have a job but on leave within the last seven days); No (no) |
| Control over own earning | The person who usually decides how to spend the respondent's earnings: Respondent alone; Respondent and husband; Husband alone/someone else |
| **Women's participation in decision-making** | |
| Own health care | Women's participation in decision-making about their healthcare: Respondent alone; Respondent and husband; Husband alone; Someone else |
| Major household purchases | The person who decided the purchase of significant household products: Respondent alone; Respondent and husband; Husband alone; Someone else |
| Child health care | Women's participation in taking decisions regarding child care: Respondent alone; Respondent and husband; Husband alone; Someone else |
| Visit family or relatives | Women's participation in decision-making regarding visiting family or relatives: Respondent alone; Respondent and husband; Husband alone; Someone else |
| Women's acceptance as justifying wife-beating | Justifying women's acceptance attitudes towards wife-beating by their husband for the following five reasons (burning food; arguing with husband; going out without telling husband; neglecting the children, and refusing to have sex with husband) and grouped as: Not justified; Any of the five reasons |

Bivariable multilevel mixed-effect logistic regression analysis was performed to estimate the direct effect of women empowerment indicators on outcomes. The results are presented as crude/unadjusted odds ratio (UOR) with a 95% confidence interval. Finally, the multivariable multilevel mixed-effect model was performed to estimate adjusted odds ratios (AOR) with 95% confidence intervals while adjusting the potential confounders selected from the conceptual framework. The significance level was considered as $p < 0.05$. The analysis was performed using Stata 15 software (Stata Corp. 2013. Stata Statistical Software: Release 13. College Station, TX: Stata Corp LP.)

## Results

Table 3 shows the symptoms of genital sore and abnormal genital discharge experienced by 16,858 currently married women. We found that the proportion of women who reported signs of genital sores and abnormal genital discharge was 5.59% and 10.84%, respectively. The

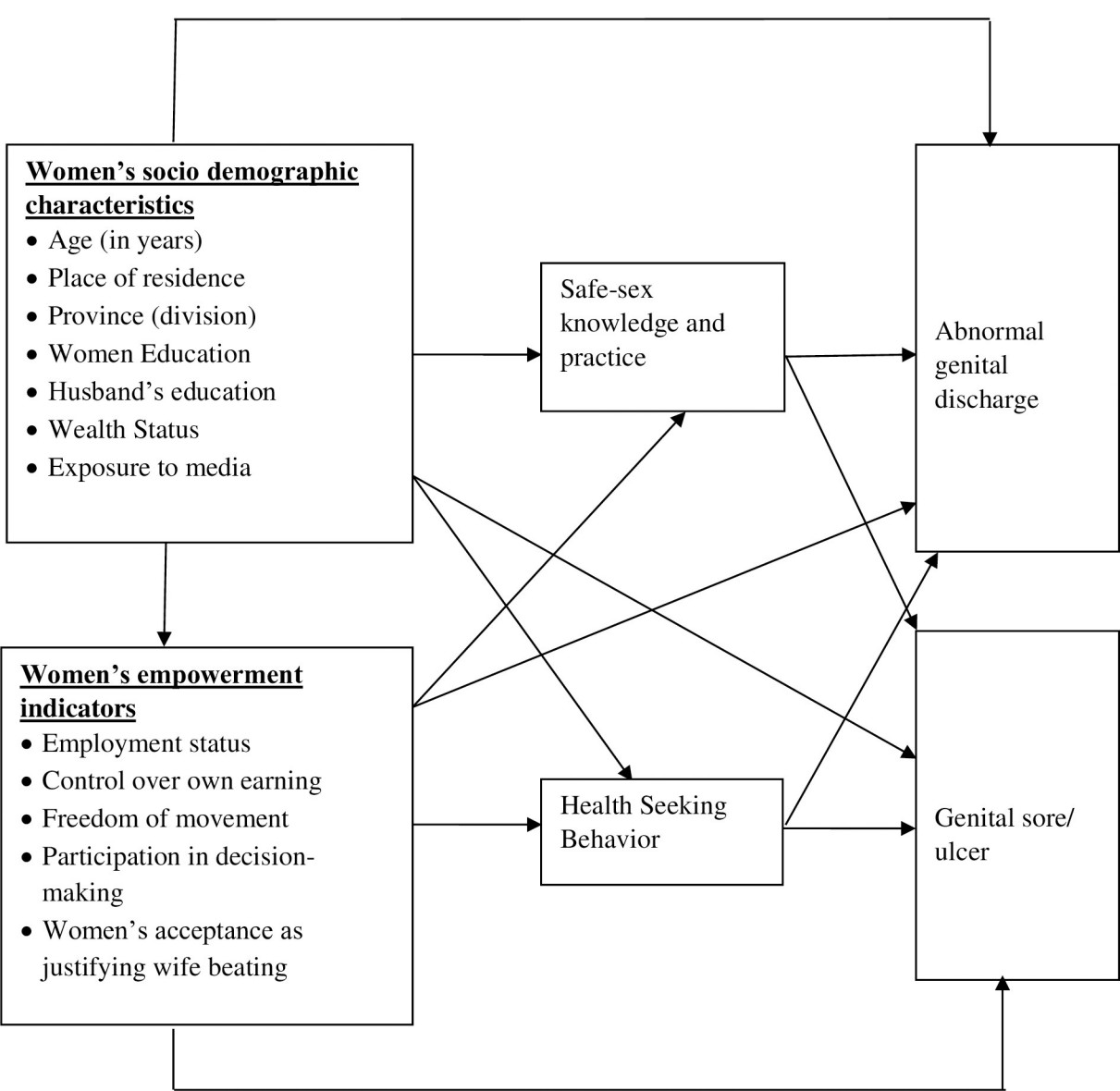

**Fig 2. The conceptual framework illustrating the causal pathway between women empowerment indicators and outcome of interest.**

average age among study samples was 30.51 years, where the average ages of women with genital sores and abnormal genital discharge symptoms were 30.58 and 30.10, respectively. We also found that higher proportions of women who reported genital sores originated from the Dhaka division (25.27%). Similarly, among those who reported signs of abnormal genital discharge, 36.99% completed their secondary education, 22.41% came from a household of median wealth, 75.04% lived in the rural area, and 32.81% were from the Dhaka division. Table 3 also shows that participating in various household decisions and acceptance towards wife-beating were significantly associated with self-reported symptoms of genital sore and abnormal genital discharge. The highest percentage of women who experienced signs of genital sore relied on their husbands' decisions on various aspects of their lives, including their healthcare (43.78%), purchasing significant household products (47.69%), child healthcare (51.17%) and visiting family or relatives (47.75%). Similarly, the percentages for the women

**Table 3. Distribution of women's self-reported experience of genital sore and abnormal genital discharge according to their socio-demographic characteristics and women empowerment indicators.**

| | | Currently married women | | | |
|---|---|---|---|---|---|
| Variables | Total | Genital sore | | Abnormal genital discharge | |
| | %(n) | %(n) | p-value[a] | %(n) | p-value[a] |
| **Overall** | 100.0(16858) | 100.0(943) | | 100.0(1827) | |
| **Age group (in the year)** | | | | | |
| Mean ± SD | 30.51 ± 0.09 | 30.58 ± 0.32 | | 30.10 ± 0.23 | |
| *15–19* | 11.78(1985) | 6.15(58) | | 20.9(382) | |
| *20–24* | 18.78(3165) | 22.04(208) | | 20.83(380) | |
| *25–29* | 19.28(3250) | 21.87(206) | | 11.55(211) | |
| *30–34* | 17.32(2919) | 20.43(193) | <0.001 | 10.77(197) | <0.001 |
| *35–39* | 12.77(2153) | 11.12(105) | | 6.03(110) | |
| *40–44* | 11.12(1875) | 11.48(108) | | 20.9(382) | |
| *45–49* | 8.96(1511) | 6.91(65) | | 20.83(380) | |
| **Women education** | | | | | |
| *No education* | 23.42(3949) | 20.96(198) | | 21.95(401) | |
| *Primary* | 29.16(4916) | 32.71(308) | 0.143 | 34.29(626) | 0.002 |
| *Secondary* | 38.58(6503) | 38.9(367) | | 36.99(676) | |
| *Higher* | 8.84(1490) | 7.44(70) | | 6.76(123) | |
| **Husband education** | | | | | |
| *No education* | 27.96(4712) | 27.65(261) | | 29.31(535) | |
| *Primary* | 27.76(4680) | 30.09(284) | 0.524 | 30.73(561) | 0.002 |
| *Secondary* | 30.17(5085) | 29.7(280) | | 28.91(528) | |
| *Higher* | 14.11(2379) | 12.57(118) | | 11.06(202) | |
| **Wealth index** | | | | | |
| *Poorest* | 18.37(3097) | 17.21(162) | | 19.13(349) | |
| *Poorer* | 19.12(3223) | 22.75(214) | | 21.68(396) | |
| *Middle* | 20.14(3395) | 21.2(200) | 0.136 | 22.41(409) | 0.002 |
| *Richer* | 21.1(3557) | 19.61(185) | | 19.88(363) | |
| *Richest* | 21.28(3587) | 19.24(181) | | 16.9(309) | |
| **Place of residence** | | | | | |
| *Urban* | 27.93(4709) | 25.7(243) | | 24.96(456) | |
| *Rural* | 72.07(12149) | 74.3(700) | 0.227 | 75.04(1370) | 0.039 |
| **Division** | | | | | |
| *Barisal* | 6.23(1051) | 9.75(92) | | 7.16(131) | |
| *Chittagong* | 18.52(3122) | 25.04(236) | | 17.12(313) | |
| *Dhaka* | 34.74(5857) | 25.27(238) | | 32.81(599) | |
| *Khulna* | 10.25(1729) | 11.45(108) | <0.001 | 12.86(235) | 0.043 |
| *Rajshahi* | 11.91(2007) | 13.26(125) | | 13.35(244) | |
| *Rangpur* | 11.54(1946) | 8.6(81) | | 10.46(191) | |
| *Sylhet* | 6.8(1147) | 6.63(63) | | 6.23(114) | |
| **Exposure to media** | | | | | |
| *No exposure* | 36.99(6235) | 36.25(342) | | 37.42(683) | |
| *Exposure to any 2 media* | 61.54(10375) | 62.63(590) | 0.790 | 61.73(1127) | 0.169 |
| *Exposure to all 3 media* | 1.47(247) | 1.12(11) | | 0.86(16) | |
| **Women empowerment indicators** | | | | | |
| **Employment Status** | | | | | |
| *No* | 65.69(11072) | 58.61(553) | <0.001 | 62.08(1134) | 0.01 |

*(Continued)*

**Table 3.** (Continued)

| Variables | Total | Currently married women | | | |
|---|---|---|---|---|---|
| | | Genital sore | | Abnormal genital discharge | |
| | %(n) | %(n) | p-value[a] | %(n) | p-value[a] |
| *Yes* | 34.31(5784) | 41.39(390) | | 37.92(693) | |
| **Control over their own earning** | | | | | |
| *Respondent alone* | 32.02(1668) | 34.14(121) | | 33.95(217) | |
| *Respondent and husband* | 54.13(2819) | 49.33(175) | 0.21 | 51.25(328) | 0.518 |
| *Husband alone/someone else* | 13.85(721) | 16.53(59) | | 14.8(95) | |
| | | | | **Women's participation in decision-making** | |
| **Own health care** | | | | | |
| *Respondent alone* | 14.12(2381) | 16.47(155) | | 17.02(311) | |
| *Respondent and husband* | 50.73(8550) | 43.78(413) | 0.002 | 43.38(792) | <0.001 |
| *Husband alone/someone else* | 35.15(5925) | 39.76(375) | | 39.6(723) | |
| **Major household purchases** | | | | | |
| *Respondent alone* | 8.3(1399) | 10.67(101) | | 11.54(211) | |
| *Respondent and husband* | 52.98(8930) | 47.69(450) | 0.005 | 45.9(838) | <0.001 |
| *Husband alone/someone else* | 38.72(6528) | 41.64(393) | | 42.57(777) | |
| **Child health care** | | | | | |
| *Respondent alone* | 16.35(2732) | 16.34(152) | | 18.12(329) | |
| *Respondent and husband* | 54.17(9054) | 51.17(477) | 0.148 | 50.56(917) | 0.106 |
| *Husband alone/someone else* | 29.48(4928) | 32.5(303) | | 31.32(568) | |
| **Visit family or relatives** | | | | | |
| *Respondent alone* | 9.84(1659) | 11.79(111) | | 11.32(207) | |
| *Respondent and husband* | 52.88(8910) | 47.75(450) | 0.017 | 45.08(823) | <0.001 |
| *Husband alone/someone else* | 37.28(6281) | 40.46(381) | | 43.6(796) | |
| **Women's acceptance as justifying wife-beating** | | | | | |
| *Not justified* | 72.62(12242) | 67.31(634) | | 69.17(1263) | |
| *Any of five reasons* | 27.38(4615) | 32.69(308) | 0.012 | 30.83(563) | 0.008 |

[a] p-values were calculated using the Pearson Chi-square test.

Note: As a result of missing values, the total may not equal 100.0 percent.

who reported abnormal genital discharge symptoms were 43.38%, 45.9% and 45.08%, respectively. 67% of women with symptoms of abnormal genital discharge agreed that wife-beating was not justified for any reason whereas, 67% of women who reported genital sores decided that wife-beating was not justified for any reason (Table 3).

## Association between women empowerment indicators and abnormal genital discharge and genital sores

Table 4 explores the association between women's empowerment indicators and their experience of genital sores and abnormal genital discharge symptoms. In the empty model, the probability of women's self-reporting genital sores and abnormal genital discharge varied significantly according to the clustering of the PSUs (genital sore, $\sigma^2 = 0.43$, 95% CI = 0.31–0.58; abnormal genital discharge, $\sigma^2 = 0.10$, 95% CI = 0.23–0.40). The ICC in the empty model indicated that differences between the clusters accounted for 11.00% and 8.00% of the total variance in women's genital sores and abnormal genital discharge, respectively.

**Table 4. Bivariable and multivariable multilevel mixed-effect logistic regression analysis to explore the association between women empowerment indicators and women's self-reported experience of genital sore and genital discharge in the past 12 months of the survey, 2014 Bangladesh.**

| Women Empowerment indicators | Genital sore | | Abnormal genital discharge | |
|---|---|---|---|---|
| | UOR[a] (95% CI) | AOR[b] (95% CI) | UOR[a] (95% CI) | AOR[b] (95% CI) |
| **Employment status** | | | | |
| *Yes* | 1.37*** (1.19–1.58) | 1.36*** (1.18–1.57) | 1.15** (1.03–1.30) | 1.14** (1.02–1.27) |
| *No* | Reference | Reference | Reference | Reference |
| **Control over their own earning** | | | | |
| *Respondent alone* | 0.91 (0.64–1.29) | 0.92 (0.65–1.31) | 0.96 (0.73–1.27) | 0.98 (0.74–1.30) |
| *Respondent and husband* | 0.78 (0.57–1.08) | 0.80 (0.57–1.10) | 0.83 (0.64–1.08) | 0.85 (0.66–1.11) |
| *Husband alone/someone else* | Reference | Reference | Reference | Reference |
| **Women's participation in decision-making** | | | | |
| **Own health care** | | | | |
| *Respondent alone* | 1.05 (0.86–1.28) | 1.04 (0.85–1.27) | 1.11 (0.95–1.28) | 1.14 (0.98–1.32) |
| *Respondent and husband* | 0.79*** (0.67–0.90) | 0.78*** (0.67–0.90) | 0.74*** (0.66–0.82) | 0.75*** (0.67–0.84) |
| *Husband alone/someone else* | Reference | Reference | Reference | Reference |
| **Major household purchases** | | | | |
| *Respondent alone* | 1.24 (0.98–1.57) | 1.22 (0.97–1.55) | 1.36*** (1.15–1.61) | 1.39*** (1.17–1.65) |
| *Respondent and husband* | 0.86** (0.75–0.99) | 0.86 (0.74–1.00) | 0.78*** (0.70–0.87) | 0.79*** (0.71–0.88) |
| *Husband alone/someone else* | Reference | Reference | Reference | Reference |
| **Child health care** | | | | |
| *Respondent alone* | 1.97 (0.76–1.15) | 0.91 (0.74–1.13) | 1.07 (0.92–1.24) | 1.09 (0.93–1.27) |
| *Respondent and husband* | 0.88 (0.75–1.02) | 0.87 (0.74–1.01) | 0.88** (0.78–0.98) | 0.89 (0.79–1.00) |
| *Husband alone/someone else* | Reference | Reference | Reference | Reference |
| **Visit to family or relatives** | | | | |
| *Respondent alone* | 1.14 (0.91–1.43) | 1.12 (0.89–1.42) | 1.01 (0.85–1.19) | 1.03 (0.87–1.23) |
| *Respondent and husband* | 0.86 (0.75–1.00) | 0.86 (0.74–1.00) | 0.71*** (0.63–0.79) | 0.72*** (0.64–0.80) |
| *Husband alone/someone else* | Reference | Reference | Reference | Reference |
| **Women's acceptance as justifying wife-beating** | | | | |
| *Not justified* | Reference | Reference | Reference | Reference |
| *Any of five reasons* | 1.28** (1.10–1.47) | 1.26** (1.08–1.46) | 1.22** (1.09–1.37) | 1.19** (1.07–1.33) |
| ***Random effect results*** | | **Null model** | | **Null Model** |
| *PSU variance (95% CI)* | | 0.43 (0.31–0.58) | | 0.10 (0.23–0.40) |
| *ICC* | | 0.11 | | 0.08 |
| *LR Test* | | $\chi^2$ = 115.66, p<0.001 | | $\chi^2$ = 179.87, p<0.001 |

**p-value < 0.05.

***p-value < 0.001.

[a]UOR represents unadjusted odd ratios which were calculated using bivariable binary logistic regression; CI indicates confidence interval

[b]AOR represents adjusted odd ratios which were calculated using multivariable binary logistic regression where variables used for adjustment: age group, women education, husband's education, wealth index, division, exposure to media, place of residence.

Three indicators were significantly associated with genital sores: women's employment status, participation in health care decision-making and women's acceptance of wife-beating. We found that employed women were 1.36 times more likely to report genital sores compared to unemployed women. In addition, women who were able to make healthcare decisions with their husbands were 22% less likely to report genital sores (AOR = 0.78) compared to women in the counter reference category. Moreover, women who accepted wife-beating for any of the five listed justified reasons had a higher likelihood of reporting genital sores (AOR = 1.26) (Table 4).

Five indicators were also found to be significantly associated with abnormal genital discharge, such as: women's employment status, their involvement in decision-making about their healthcare, major household purchases, visits to family or relatives and their acceptance of wife-beating. Women were 1.14 times (AOR = 1.14) more likely to experience symptoms of genital discharge than unemployed women. Conversely, the odds of having genital discharge symptoms were lowest among women who relied on their husbands to make decisions regarding significant household purchases, their own healthcare, visiting family or relatives (i.e., 0.79, 0.75, 0.72, respectively compared to the reference category). In addition, women who accepted wife-beating for any of the five mentioned reasons were 1.19 times more likely to report abnormal genital discharge compared to the reference sample. Besides, women's control over their earnings, involvement in significant household purchases, child healthcare, and decision-making about visiting family members or relatives were not significantly associated with self-reported genital sores. Likewise, we found no significant association between women's control over their earnings or child health care decision-making and women who experienced abnormal genital discharge symptoms (Table 4).

## Discussion

In this study, 5.59% and 10.84% of the respondents reported genital sores and abnormal genital discharge, respectively. We found a significant relationship between the indicators of women empowerment and their likelihood of reporting symptoms of STIs such as genital sores or abnormal genital discharge. Women who actively participated in joint decision-making with their husbands or partners regarding their family's healthcare were significantly less likely to report symptoms of STIs. Joint decision-making about visiting family or making significant purchases also lessened their likelihood of reporting symptoms of STIs. Notably, we discovered that women with a history of employment were more likely to report symptoms of STIs. Likewise, women who justified their husbands' wife beatings under various circumstances were more likely to report symptoms of STI. These findings generate crucial insights that warrant further deliberation, particularly within the patriarchal socio-cultural context in Bangladesh.

The benefit of joint decision-making calls into question the individualistic autonomy model, which portrays women as independent and autonomous characters who are also shouldered with the responsibility to make the decision [36–38]. However, within the South Asian social framework, where the women's status is inherently tied to men by emotional interdependence, the autonomy framework may not be contextually appropriate [37, 38]. Rather, receiving support from the husband fosters better service utilisation and uptake of necessary healthcare among women [39]. It is highly likely that joint discussions and deliberations about a decision are more likely to produce better outcomes for the women [39]. Previous studies in Bangladesh also suggest that joint participation in making household decisions may precipitate better results than independent decision-making, thus undermining the concept of women's unilateral decision-making [37]. These findings support a solid argument for an environment that enables couples to consult, negotiate and overcome conflicting preferences and goals about family issues.

However, women's involvement in joint decision-making is not a smooth process. Women's educational status and the socio-economic status of their families of origin are conceptualised as prerequisites for their decision-making abilities [40]. Women with higher levels of education are more likely to be involved in decision-making with their husband. Her involvement in income-generating activities further potentiates her capacity in the household [41]. Women with greater household decision-making powers are more likely to uptake sexual and

reproductive health services than those with less power [42–44]. In the egalitarian setting of Tamil Nadu in India, women's economic activity positively influenced decision making, an unlikely phenomenon in the gender-conservative context of several locales of India [45]. In recent times, women's employment in garments and microfinance-based development programs in Bangladesh has considerably increased the scope for women empowerment. Because of these particular initiatives, women are now gaining more control over their sexual and reproductive life by delaying marriage and childbirth, for instance [46, 47]. All these factors significantly contribute to a gradual shift in gender norms and nurture greater independence and economic power in the family for women, which ultimately enhances the uptake of certain sexual and reproductive health services.

Although the involvement of women in income-generating activities is hypothesised as a critical factor in empowerment, our analysis underscored a few crucial aspects. For example, we found that women empowerment during the past year resulted in the reporting of more genital sores and abnormal genital discharges. Global evidence suggests that being employed and spending longer working hours, along with substance abuse, peer pressure, and inadequate guardianship, may increase their propensity towards unsafe sexual exposure, especially for teenage and young women [48–50]. If occupational safety is not ensured, women may become isolated and vulnerable at their workplace, which could exacerbate the possibility of sexual exposure [49]. Therefore, policy planners need to ensure a safe working environment for women if they want to continue engaging them in income-generating activities.

In addition, our study presented that women who justified reasons for wife-beating attitudes were more likely to report both abnormal genital discharge and genital sores. The perception of IPV as a "similar notion" resonated in other LMIC like Nigeria and Zambia, which embody similar patriarchal norms [51, 52]. Women living in an environment that normalises wife-beating were more likely to be younger, less educated and from a lower socio-economic background, as they blamed themselves for such incidents, thus further enabling perpetrators to continue this behavior [53, 54]. These findings add to the body of knowledge that shows the effects of IPV on Bangladeshi women, including unwanted pregnancies, induced abortions, miscarriage and stillbirth [55]. IPV needs to be acknowledged as a social malignancy that insidiously affects women's physical and emotional welfare, thereby underpinning women empowerment initiatives as an evidence-based solution [22]. Thus, policymakers must carefully design programs that empower women through education and microfinance, encourage women to speak out about IPV and establish social and legal structures that protect women in such situations and strengthen the IPV prevention paradigm.

In a predominantly Muslim country like Bangladesh, the conservative socio-economic environment is not conducive to public discussions about sexual health issues and behaviours. Furthermore, the lack of appropriate knowledge about STIs prevention and transmission has elevated the risk of STIs transmission. Due to their lack of awareness about STIs and their associated harms, it is unlikely that women can discuss these issues with their husbands, thus limiting their scope for safer sex negotiation [56]. Women empowerment initiatives would provide women with greater economic autonomy to reject men, multiple partners, insist on safer sex and frequently discuss HIV/STIs vulnerabilities with male partners [57–59]. Thus, economic empowerment initiatives need to consider STIs prevention efforts to foster their economic empowerment while providing them with resources to protect themselves from STIs.

Our analysis was based on data collected in a national survey with high response rates. To ascertain national representation, the multistage sampling procedure used in the survey was adjusted using sampling weights. The variables in the study were related to socio-demographic characteristics and women empowerment. Models were developed to illustrate the association between women empowerment and the symptoms of STI, genital sores, and foul-smelling

genital discharge, which could aid policymakers and activists in designing interventions to reduce symptoms of STI through women empowerment initiatives. However, this study presented some limitations. Even though a later survey was conducted in 2017 and published, the dataset did not contain any STIs-related information. Since these data were not available, a similar analysis cannot be demonstrated. Moreover, due to the sensitive and stigmatised nature of STIs and sexual health, respondents may not completely disclose their sexual health information, thus leading to social desirability bias or self-reporting bias. Strictly maintaining anonymity and confidentiality at the time of data collection minimises the occurrence of these biases, thus preventing inaccurate estimates of association or underestimation of risk parameters. In addition, according to the standard definition of STIs, laboratory diagnostics are required, along with clinical signs and symptoms. However, as BDHS is a population-based survey, there were no provisions for confirmatory diagnostics for participants with self-reported symptoms of STI. Therefore, to determine STIs status, further studies are needed to utilise clinical diagnostic approaches, which would elicit a more in-depth understanding of the association between STIs and women empowerment among currently married women.

## Conclusion

In this study, we have shown that women's joint decision-making can be considered a reliable marker of women empowerment in Muslim societies such as Bangladesh, which normalises male dominance. This collective decision-making capacity was eventually linked to lower reported symptoms of STIs, thus highlighting the benefits of egalitarian gender norms on women's sexual and reproductive health. Women need to attain equitable empowerment to negotiate participation in joint decision-making and exercise their rights in their families and society. Thus, it is integral for women to attain education, health awareness and economic stability. Moreover, additional mechanisms need to be explored to encourage couple communication strategies to improve relationship dynamics. Policy decisions should consider contextual and evidence-based findings to formulate acceptable gender policies for facilitating a better life for women.

## Acknowledgments

We want to acknowledge the contribution of the Bangladesh Demographic and Health Survey (BDHS), National Institute of Population Research and Training (NIPORT/Bangladesh), MEASURE DHS as well as ICF International teams (of USA) for their efforts to collect and give permission to use the Bangladesh Demographic Health Survey, 2014 data. We want to acknowledge the contribution of International Diarrhoeal Diseases Research, Bangladesh (icddr,b) to permit writing this research article. We also would like to acknowledge the donors providing unrestricted support to icddr,b's research efforts. icddr,b is grateful to the governments of Bangladesh, Canada, Sweden and the UK for providing core/unrestricted support. Also, we would like to express our gratitude to Samira Dishti Ifran for her support in editing and reviewing the manuscript.

## Author Contributions

**Conceptualization:** Md Abdullah Al Jubayer Biswas.

**Data curation:** Md Abdullah Al Jubayer Biswas.

**Formal analysis:** Md Abdullah Al Jubayer Biswas.

**Investigation:** Md Abdullah Al Jubayer Biswas.

**Methodology:** Md Abdullah Al Jubayer Biswas, Mohammad Abdullah Kafi.

**Software:** Md Abdullah Al Jubayer Biswas.

**Supervision:** Mohammad Abdullah Kafi, Sharful Islam Khan.

**Visualization:** Md Abdullah Al Jubayer Biswas.

**Writing – original draft:** Md Abdullah Al Jubayer Biswas, Mohammad Abdullah Kafi, Muhammad Manwar Morshed Hemel.

**Writing – review & editing:** Md Abdullah Al Jubayer Biswas, Mohammad Abdullah Kafi, Muhammad Manwar Morshed Hemel, Mondar Maruf Moin Ahmed, Sharful Islam Khan.

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
