## [Decision Letter · Decision Letter 0]

19 Apr 2021

PONE-D-21-00487

Women empowerment and sexually transmitted infections: Evidence from Bangladesh demographic and health survey 2014

PLOS ONE

Dear Dr. Khan,

Thank you for submitting your manuscript to PLOS ONE. I have now received the review reports from the two reviewers. I have also read the manuscript with interest. After careful consideration, we feel that it has merit but does not fully meet PLOS ONE’s publication criteria as it currently stands. Therefore, we invite you to submit a revised version of the manuscript that addresses the points raised during the review process. Please see my comments below, along with the reviewer’s reports.

Line 113 mentioned that “All the independent variables are shown in Table 1”. It will be Table 2.The authors have not constructed an empowerment scale/index. They used the indicators as a separate variable.The division has been written as ‘Province.’The authors have presented a conceptual framework where knowledge about safe sexual behavior and treatment-seeking behavior for STI symptoms have been mentioned. However, I did not see any such variable in the list of explanatory variables and the results.The reported sample 16858 does not match with the DHS report. The same goes for the proportion of women who reported having signs of genital sores and abnormal genital discharge. Please check the original report.In table 3, the authors have reported findings of the variable “Control over their ow earnings”. However, the ‘n’ is different from the total. However, no explanation has been provided.Line 162 mentioned, “***Impact of women empowerment on abnormal genital discharge and genital sores”. ***Please consider revisiting the use of “impact” as the current study design does not allow the authors to see the impact.

We look forward to receiving your revised manuscript.

Kind regards,

Mohammad Bellal Hossain, PhD

Academic Editor

PLOS ONE

Journal Requirements:

'The funders had no role in study design, data collection and analysis, decision to publish, or preparation of the manuscript'

Please include your amended statements within your cover letter; we will change the online submission form on your behalf.v

4. Please include a caption for figure 1.

Additional Editor Comments (if provided):

Reviewers' comments:

Reviewer's Responses to Questions

**Comments to the Author**

1. Is the manuscript technically sound, and do the data support the conclusions?

Reviewer #1: Partly

Reviewer #2: Yes

2. Has the statistical analysis been performed appropriately and rigorously? 

Reviewer #1: Yes

Reviewer #2: No

3. Have the authors made all data underlying the findings in their manuscript fully available?

Reviewer #1: Yes

Reviewer #2: Yes

4. Is the manuscript presented in an intelligible fashion and written in standard English?

Reviewer #1: No

Reviewer #2: Yes

5. Review Comments to the Author

Reviewer #1: On Abstract

- The second sentence of your abstract is not clear. What poses difficulties and what remains untreated? I think you should read it again and try to make it clear. It can be rephrased as … many women remain untreated

- Please write the word cross-tab in full

- Change the and in “All the analysis was adjusted for multistage sampling design and cluster weight.” to …. using cluster weight

- The sentence that presents multivariate results is not clear. Please read it again and present it clearly. I think it should be presented as

Multivariate analysis revealed that women’s lower reporting of STIs symptoms was significantly…..

- The figures in the abstract should be presented with one decimal place.

On Manuscript

- Your introduction needs thorough editing. Many of the sentences lack joining words and the manuscript is full of grammatical errors.

- You should have some literature that connects other selected variables to real studies. The introduction is silent about these connections yet they are also study vaiables.

- The first sentence of the introduction is not clear. Seems you are joining two parts that are disconnected.

- Line 62 misses … in developing countries.

- Line 62 It has also been reported that there is higher HIV transmission among reproductive-age women

- Line 65 misses … and genital herpes

- Lines 90-93 are confusing about the dataset used in this study. You state that the 2017 BDHS did not have STI/HIV data but then conclude that you used the 2017 BDHS. I think line 92 should read …, BDHS 2014 …

-

- Since you have two outcomes, the subsection in line 105 should be outcome variables

- The word “referent” in the multivariate table should be changed to “reference”

- In the discussion, the sentence on line 215 should end on independently otherwise, the addition renders it confusing.

- Line 220-223 is very questionable for women of reproductive age. Is this in the context of your community. It may only apply to the teenagers and young women. If so, please indicate this clearly. Otherwise many of the women of reproductive age may not be under any guardians!

- In the conclusion, I think the issue of joint decision making can also be facilitated by encouraging couple communication. If it is applicable in the Bangladesh Context, please include it.

Reviewer #2: I have gone through the MS carefully. The author(s) have tried to identify the impact of women’s empowerment on sexually transmitted infections STIs). However, my observations are as follows:

Abstract:

The abstract is well written.

Introduction:

Introductory section is very short. The authors urged that (P. 9, L. 60-62) abnormal genital discharge is a common occurrence caused mostly by Chlamydia trachomatis, Neisseria gonorrhoeae, Trichomonas vaginalis and is associated with a higher risk of HIV transmission precisely developing countries. Citing of some findings from developing countries which showed that having STIs in a woman increases HIV would increase the quality of the MS. Are those infections limited to HIV, or some other diseases like cancer?

Materials and Methods:

Some following issues are not clear in this section:

(i) I am not sure when the MS was submitted. The latest BDHS data sets conducted in 2017-18 are released by the end of November or in early of December, 2020. If so, why the authors have not used most recent data?

(ii) A flow chart of selection of the 16,858 from 17,886 women may improve the quality of the MS.

(iii) Since the data have been gathered through a multi-stage procedure, why di not the authors use multilevel logistic regression analysis. The empirical studies suggest that women’s empowerment are associated with clusters. I would suggest employing multilevel logistic regression analysis to quantify the unobserved effects on having any STI among married women captured by clusters.

(iv) How the missing values were managed by stata?

Discussion:

This section is well written.

Conclusion:

This section is well written.

Overall comments:

The authors have examined an important public health issues. Overall, the writing is good, well arranged, and lucid. However, some grammatical issues should be checked throughout the MS. In this study, to my opinion, the main backdrops are data and methodological issues as noted above. Hence it needs a revision. Prior to publish the MS, the above mentioned issues should be taken into consideration and clarified.

6. PLOS authors have the option to publish the peer review history of their article (what does this mean?). If published, this will include your full peer review and any attached files.

Reviewer #1: No

Reviewer #2: **Yes: **S. M. Mostafa Kamal

---

## [Author Response · Author response to Decision Letter 0]

23 Jun 2021

Response to Editor Comments

1.Line 113 mentioned that “All the independent variables are shown in Table 1”. It will be Table 2.

Response: Thank you. We have corrected it in line 144 of the unmarked manuscript file.

2.The authors have not constructed an empowerment scale/index. They used the indicators as a separate variable

Response: Thank you for highlighting this important concern, we appreciate it. It was possible to employ a women empowerment index, such as the Survey-based Women's Empowerment Index-1. However, rather than using a composite score, we were interested in investigating how different women empowerment indicators might affect women's self-reported abnormal genital discharge and genital sores differently2.

Reference: 

1. Ewerling F, Lynch JW, Victora CG, van Eerdewijk A, Tyszler M, Barros AJ. The SWPER index for women's empowerment in Africa: development and validation of an index based on survey data. The Lancet Global Health. 2017 Sep 1;5(9):e916-23.

2. Mainuddin A, Begum HA, Rawal LB, Islam A, Islam SS. Women empowerment and its relation with health seeking behavior in Bangladesh. J Family Reprod Health. 2015;9(2):65.

3.The division has been written as ‘Province.’

Response: Thank you. In the context of Bangladesh, the term “province” is not officially used. The word “Division” is an officially accepted term. We have corrected it in Table 2, Line 148, 194, Table 3, Table 4 of the unmarked manuscript file (Table 2, Line 117, 145, 148)

4.The authors have presented a conceptual framework where knowledge about safe sexual behavior and treatment-seeking behavior for STI symptoms have been mentioned. However, I did not see any such variable in the list of explanatory variables and the results

Response: Thank you for your comment. Women's socio-demographic characteristics and indicators of women's empowerment have direct and indirect effects on abnormal genital discharge and genital sores. A single direct arrow indicates a direct effect. Women’s demographic characteristics and empowerment indicators indirectly also affect outcomes through safer sex knowledge and practices and health-seeking behaviour. Since we do not have data on safer sex knowledge, practices and health-seeking behaviour, our model exclusively focused on the direct effect. Now we have revised the statistical analysis section and explained it in Lines 158–169 of the unmarked manuscript file.

5.The reported sample 16858 does not match with the DHS report. The same goes for the proportion of women who reported having signs of genital sores and abnormal genital discharge. Please check the original report

Response: Many thanks for your comment. Our sample size was different from the BDHS sample size because the BDHS report [Table 3.1 (page 29) and Table 12.7 (page 184)] calculated the prevalence among ever-married women, which included married, widowed, divorced, and women who were no longer living together (sample size=17,868). However, we limited our analysis to currently married women and thus used a sample size of 16,858 women. We referenced it in the study design and sampling technique section, line 130-132 of the unmarked manuscript file.

6.In table 3, the authors have reported findings of the variable “Control over their ow earnings”. However, the ‘n’ is different from the total. However, no explanation has been provided.

Response: We appreciate your comment. In table 3, now we have included a footnote that as a result of missing values, the total may not equal 100.0 percent.

7.Line 162 mentioned, “Impact of women empowerment on abnormal genital discharge and genital sores”. Please consider revisiting the use of “impact” as the current study design does not allow the authors to see the impact.

Response: We appreciate this concern. We have revised it to the association between women empowerment indicators and abnormal genital discharge and genital sores in line 211-212 in the unmarked manuscript file. 

 

Response to Reviewer 1 Comments

On Abstract

Comment 1: The second sentence of your abstract is not clear. What poses difficulties and what remains untreated? I think you should read it again and try to make it clear. It can be rephrased as … many women remain untreated

Response: We appreciate your suggestion. We used the term "difficulty" and "untreated" to refer to "challenges" and "undiagnosed." However, in the unmarked manuscript, we revised lines 29-33 in the previous edition and inserted them in line -23-27.

Comment 2: Please write the word cross-tab in full

Response: Thank you for the suggestion. It has been inserted in unmarked manuscript lines 28-32. In addition, we revised the materials and methods section to reflect the fact that we used multilevel mixed-effect logistic regression as suggested by reviewer 2.

Comment 3: Change the and in “All the analysis was adjusted for multistage sampling design and cluster weight.” to …. using cluster weight

Response: We appreciate the suggestion. In line 32 (lines 35-36 in the previous edition or version?) of the unmarked manuscript, we have inserted the phrase using cluster weight.

Comment 4: The sentence that presents multivariate results is not clear. Please read it again and present it clearly. I think it should be presented as Multivariate analysis revealed that women’s lower reporting of STIs symptoms was significantly…..

Response: Thanks for the suggestion. We re-read and revised lines 34–39 of the unmarked manuscript (lines 38-41 in the previous version) as per your recommendation. 

Comment 5: The figures in the abstract should be presented with one decimal place.

Response: Thank you for highlighting this. We have rounded the figures only in the abstract section. 

On Manuscript

Comment 6: Your introduction needs thorough editing. Many of the sentences lack joining words and the manuscript is full of grammatical errors.

Response: We appreciate this suggestion. We have re-read and edited the introduction section thoroughly line-by-line, attempting to correct grammatical and syntax errors throughout the manuscript. 

Comment 7: You should have some literature that connects other selected variables to real studies. The introduction is silent about these connections yet they are also study vaiables.

Response: We appreciate your recommendation. We revised the introduction section extensively, attempted to connect the study variables, and added some literature from lower-middle income countries.

Comment 8: The first sentence of the introduction is not clear. Seems you are joining two parts that are disconnected.

Response: Many thanks. We have revised this according to your suggestion. 

Comment 9: Line 62 misses … in developing countries.

Response: We appreciate your diligence in following up on this. In line 55-65 of the unmarked manuscript, we added some literature pertaining to developing countries.

Comment 10: Line 62 It has also been reported that there is higher HIV transmission among reproductive-age women

Response: Thank you, we have deleted line 62 after thoroughly rewriting the introduction.

Comment 11: Line 65 misses … and genital herpes

Response: We appreciate your feedback, and we have removed line 65 after a detailed revision of the introduction.

Comment 12: Lines 90-93 are confusing about the dataset used in this study. You state that the 2017 BDHS did not have STI/HIV data but then conclude that you used the 2017 BDHS. I think line 92 should read …, BDHS 2014 …

Response: Thank you for highlighting this. It was a typing error, and we apologize for this. Now we have corrected it in line 114-118 of the unmarked manuscript (line 90-94 in the previous edition)

Comment 13: Since you have two outcomes, the subsection in line 105 should be outcome variables

Response: Thank you for the suggestion. We have been split sentence and added a few sentences. The new edition lines ran from 136 to 140, replacing the previous edition line of 106-109.

Comment 14: The word “referent” in the multivariate table should be changed to “reference”

Response: Many thanks, we have modified this in the table as suggested. 

Comment 15: In the discussion, the sentence on line 215 should end on independently otherwise, the addition renders it confusing.

Response: Thanks for your comment. The clause in line 215 currently ends separately, and a clarification has been provided ('Also' has been omitted from the following sentence). The line is 253 in the unmarked version.

Comment 16: Line 220-223 is very questionable for women of reproductive age. Is this in the context of your community. It may only apply to the teenagers and young women. If so, please indicate this clearly. Otherwise many of the women of reproductive age may not be under any guardians!

Response: We appreciate this suggestion. The statement was made in reference to teenage and young women, which we have addressed now (Line-295). The previous version has a line number of 223.

Comment 17: In the conclusion, I think the issue of joint decision making can also be facilitated by encouraging couple communication. If it is applicable in the Bangladesh Context, please include it.

Response: Thanks for the suggestion. We agree with this, and we revised the conclusion (Line-346 in unmarked manuscript) which was Line-299 & 300 in the previous version.

Response to Reviewer 2 Comments

Introduction

Comment 1: Introductory section is very short. The authors urged that (P. 9, L. 60-62) abnormal genital discharge is a common occurrence caused mostly by Chlamydia trachomatis, Neisseria gonorrhoeae, Trichomonas vaginalis and is associated with a higher risk of HIV transmission precisely developing countries. Citing of some findings from developing countries which showed that having STIs in a woman increases HIV would increase the quality of the MS. Are those infections limited to HIV, or some other diseases like cancer?

Response: Thank you for discussing this important question. Chlamydia trachomatis, Neisseria gonorrhoeae, Trichomonas vaginalis are not only responsible for HIV propagation but also for the transmission of many other diseases such as cancer. We extensively re-read the literature and accordingly have revised the introduction section as per the reviewer’s recommendation. In addition, we provided specific literature pertaining to developing nations. The unmarked manuscript contains updated lines 49-111, while the previous edition included lines 55–87.

Materials and Methods

Comment 2: I am not sure when the MS was submitted. The latest BDHS data sets conducted in 2017-18 are released by the end of November or in early of December, 2020. If so, why the authors have not used most recent data?

Response: Thank you for highlighting this. At the time of submission (December 2020, https://dhsprogram.com/data/available-datasets.cfm), the BDHS 2017-18 data set was available. However, we were unable to find any data on STIs in the 2017-18 BDHS. Although the variables exist in the dataset, we did not find any observations for them.

Comment 3: A flow chart of selection of the 16,858 from 17,886 women may improve the quality of the MS.

Response: Thank you for your suggestion. We have included a flow chart in the TIFF file, titled "A flow chart of study population selection from the Bangladesh demographic health survey (BDHS) 2014."

Comment 4: Since the data have been gathered through a multi-stage procedure, why did not the authors use multilevel logistic regression analysis. The empirical studies suggest that women’s empowerment are associated with clusters. I would suggest employing multilevel logistic regression analysis to quantify the unobserved effects on having any STI among married women captured by clusters.

Response: We would like to thank the reviewer for raising this crucial suggestion regarding employing the multilevel logistic regression model. We previously considered the marginal model and adjusted the weight adopted from BDHS in our analysis. However, we have now re-run our analysis using multilevel mixed-effect logistic regression. Moreover, we have revised the statistical analysis section (lines 155-184 in the new edition), the results section (lines 213-240 in the new edition), and table-4 to reflect these changes. In addition, we have updated our discussion section in line with the results section.

Comment 5: How the missing values were managed by stata?

Response: Thank you for highlighting this issue. We did not employ any statistical analysis process such as missing at random or missing completely at random to manage missing values in our analysis. We conducted the analysis using the available complete data in the sample. In the BDHS report the authors did not conduct any missing value operation rather they reported if the missing value were presents. We have added a footnote to Table-3 to indicate that the total may not equal 100.0 percent due to missing values.

---

## [Decision Letter · Decision Letter 1]

26 Oct 2021

PONE-D-21-00487R1Women empowerment and sexually transmitted infections: Evidence from Bangladesh demographic and health survey 2014PLOS ONE

Dear Dr. Khan,

Thank you for submitting your manuscript to PLOS ONE. After careful consideration, we feel that it has merit but does not fully meet PLOS ONE’s publication criteria as it currently stands. Therefore, we invite you to submit a revised version of the manuscript that addresses the points raised during the review process.

We look forward to receiving your revised manuscript.

Kind regards,

Mohammad Bellal Hossain

Academic Editor

PLOS ONE

Journal Requirements:

Reviewers' comments:

Reviewer's Responses to Questions

**Comments to the Author**

1. If the authors have adequately addressed your comments raised in a previous round of review and you feel that this manuscript is now acceptable for publication, you may indicate that here to bypass the “Comments to the Author” section, enter your conflict of interest statement in the “Confidential to Editor” section, and submit your "Accept" recommendation.

Reviewer #1: All comments have been addressed

2. Is the manuscript technically sound, and do the data support the conclusions?

Reviewer #1: Yes

3. Has the statistical analysis been performed appropriately and rigorously? 

Reviewer #1: Yes

4. Have the authors made all data underlying the findings in their manuscript fully available?

Reviewer #1: Yes

5. Is the manuscript presented in an intelligible fashion and written in standard English?

Reviewer #1: No

6. Review Comments to the Author

Reviewer #1: I congratulate the authors on making the manuscript better, I however have these comments.

1. Line 34 lacks and between cross-tab and logistic regression

2. Line 37, express figures in the same format. I see that some of them presented to one decimal place yet others are presented to two decimal places. Make this consistent throughout the manuscript

3. Line 57 should be women’s well-being

4. Line 62 : reported to lead to

5. Line 105: Outcome variables

6. In table 1, I think you refer to recoded not recorded

7. In table 3, control and decision making variables are not presented well. The names of the categories are cut – some words are below the lines. I think you can consider having the table in a smaller font so that the names of the variables and their categories appear in whole.

8. In table 4, the word is reference not referent

9. Line 213, use have instead of has

10. Line 214, in not from

11. Line 239, delete help

You need a final English edit for the manuscript

7. PLOS authors have the option to publish the peer review history of their article (what does this mean?). If published, this will include your full peer review and any attached files.

Reviewer #1: No

---

## [Author Response · Author response to Decision Letter 1]

7 Nov 2021

Response to Journal Requirements

Response: Thank you very much. It is really appreciated. We modified certain references during the Round 1 revision. The following is a list of changed references.

Retracted references (lines 520-642 in track changed version)

1. Fabrizio S, Kolovich L, Newiak M, Agarwal A, Yin RJ. Pursuing Women's Economic Empowerment. (IMF Web site): Accessed July 13, 2019. https://www.imf.org/en/Publications/Policy-Papers/Issues/2018/05/31/pp053118pursuing-womens-economic-empowerment. , 2019.

2. Gibson CJ, Huang AJ, McCaw B, Subak LL, Thom DH, Van Den Eeden SK. Associations of Intimate Partner Violence, Sexual Assault, and Posttraumatic Stress Disorder With Menopause Symptoms Among Midlife and Older Women. JAMA Intern Med. 2019;179(1):80-7. Epub 2018/11/20. doi: 10.1001/jamainternmed.2018.5233. PubMed PMID: 30453319.

3. Treves-Kagan S, El Ayadi AM, Morris JL, Graham LM, Grignon JS, Ntswane L, et al. Sexual and Physical Violence in Childhood Is Associated With Adult Intimate Partner Violence and Nonpartner Sexual Violence in a Representative Sample of Rural South African Men and Women. J Interpers Violence. 2019:886260519827661. Epub 2019/02/09. doi: 10.1177/0886260519827661. PubMed PMID: 30735091

5. Prakash R, Singh A, Pathak PK, Parasuraman S. Early marriage, poor reproductive health status of mother and child well-being in India. J Fam Plann Reprod Health Care. 2011;37(3):136-45. Epub 2011/06/02. doi: 10.1136/jfprhc-2011-0080. PubMed PMID: 21628349

6. Kaljee LM, Green M, Riel R, Lerdboon P, Tho le H, Thoa le TK, et al. Sexual stigma, sexual behaviors, and abstinence among Vietnamese adolescents: implications for risk and protective behaviors for HIV, sexually transmitted infections, and unwanted pregnancy. J Assoc Nurses AIDS Care. 2007;18(2):48-59. Epub 2007/04/04. doi: 10.1016/j.jana.2007.01.003. PubMed PMID: 17403496; PubMed Central PMCID: PMCPmc2063998

9. Ilankoon IMPS, Goonewardena CSE, Perera PPR, Fernandopulle R. Vaginal Discharge Women's Health Seeking Behaviours And Cultural Practices. 2015.

10. Gomes CMM, Giraldo PC, Gomes FdAM, Amaral R, Passos MRL, Gonçalves AKdS. Genital ulcers in women: clinical, microbiologic and histopathologic characteristics. Brazilian Journal of Infectious Diseases. Apr. 2007;11(2).

11. Sheffield JS, Wendel GD, Jr., McIntire DD, Norgard MV. Effect of genital ulcer disease on HIV-1 coreceptor expression in the female genital tract. The Journal of infectious diseases. 2007;196(10):1509-16. Epub 2007/11/17. doi: 10.1086/522518. PubMed PMID: 18008231.

24. Hebling EM, Guimaraes IR. Women and AIDS: gender relations and condom use with steady partners. Cad Saude Publica. 2004;20(5):1211-8. Epub 2004/10/16. doi: /S0102-311x2004000500014. PubMed PMID: 15486663.

25. MacPherson EE, Sadalaki J, Njoloma M, Nyongopa V, Nkhwazi L, Mwapasa V, et al. Transactional sex and HIV: understanding the gendered structural drivers of HIV in fishing communities in Southern Malawi. J Int AIDS Soc. 2012;15 Suppl 1:1-9. Epub 2012/06/22. doi: 10.7448/ias.15.3.17364. PubMed PMID: 22713352; PubMed Central PMCID: PMCPmc3499929.

26. Bloom SS, Wypij D, Gupta MD. Dimensions of women’s autonomy and the influence on maternal health care utilization in a north Indian city. Demography. 2001;38(1):67-78.

33. Haque SE, Rahman M, Mostofa MG, Zahan MS. Reproductive health care utilization among young mothers in Bangladesh: does autonomy matter? Women's Health Issues. 2012;22(2):e171-e80

Added references (lines 654-807 in track changed version)

1. Korenromp EL, Rowley J, Alonso M, Mello MB, Wijesooriya NS, Mahiane SG, et al. Global burden of maternal and congenital syphilis and associated adverse birth outcomes-Estimates for 2016 and progress since 2012. PLoS One. 2019;14(2):e0211720. Epub 2019/02/28. doi: 10.1371/journal.pone.0211720. PubMed PMID: 30811406; PubMed Central PMCID: PMCPMC6392238.

2. Prevention CfDCa. Syphilis - CDC Fact Sheet. Available from: https://www.cdc.gov/std/syphilis/stdfact-syphilis.htm.

4. Organisation WH. Sexually transmitted infections (STIs) February 28 2019. Available from: https://www.who.int/news-room/fact-sheets/detail/sexually-transmitted-infections-(stis).

5. Wand H, Ramjee G. Assessing and evaluating the combined impact of behavioural and biological risk factors for HIV seroconversion in a cohort of South African women. AIDS care. 2012;24(9):1155-62.

6. Davey DJ, Shull H, Billings J, Wang D, Adachi K, Klausner J. Prevalence of curable sexually transmitted infections in pregnant women in low-and middle-income countries from 2010 to 2015: a systematic review. Sexually transmitted diseases. 2016;43(7):450.

7. Chesson HW, Mayaud P, Aral SO. Sexually transmitted infections: impact and cost-effectiveness of prevention. 2017.

8. Mayaud P, Mabey D. Approaches to the control of sexually transmitted infections in developing countries: old problems and modern challenges. Sexually transmitted infections. 2004;80(3):174-82.

9. Alomair N, Alageel S, Davies N, Bailey JV. Sexually transmitted infection knowledge and attitudes among Muslim women worldwide: a systematic review. Sexual and reproductive health matters. 2020;28(1):1731296.

10. Jahangir YT, Arora A, Liamputtong P, Nabi MH, Meyer SB. Provider Perspectives on Sexual Health Services Used by Bangladeshi Women with mHealth Digital Approach: A Qualitative Study. International Journal of Environmental Research and Public Health. 2020;17(17):6195

14. Kabir A, Rashid MM, Hossain K, Khan A, Sikder SS, Gidding HF. Women’s empowerment is associated with maternal nutrition and low birth weight: Evidence from Bangladesh Demographic Health Survey. BMC women's health. 2020;20:1-12

15.James-Hawkins L, Peters C, VanderEnde K, Bardin L, Yount KM. Women’s agency and its relationship to current contraceptive use in lower-and middle-income countries: A systematic review of the literature. Global Public Health. 2018;13(7):843-58

16. Ahmed S, Creanga AA, Gillespie DG, Tsui AO. Economic status, education and empowerment: implications for maternal health service utilisation in developing countries. PloS one. 2010;5(6):e11190

18. Yaya S, Uthman OA, Ekholuenetale M, Bishwajit G. Women empowerment as an enabling factor of contraceptive use in sub-Saharan Africa: a multilevel analysis of cross-sectional surveys of 32 countries. Reproductive health. 2018;15(1):1-12

19. Afroja S, Rahman M, Islam L. Women’s Autonomy and Reproductive Healthcare-Seeking Behavior in Bangladesh: Further Analysis of the 2014 Bangladesh Demographic and Health Survey.

20. Story WT, Burgard SA. Couples’ reports of household decision-making and the utilisation of maternal health services in Bangladesh. Social science & medicine. 2012;75(12):2403-11.

21. Parvin GA, Ahsan SR, Chowdhury MR. Women empowerment performance of income generating activities supported by Rural Women Employment Creation Project (RWECP): A case study in Dumuria Thana, Bangladesh. The Journal of Geo-Environment. 2004;4(1):47-62.

22. Schuler SR, Lenzi R, Badal SH, Bates LM. Women’s empowerment as a protective factor against intimate partner violence in Bangladesh: a qualitative exploration of the process and limitations of its influence. Violence against women. 2017;23(9):1100-21.

23. Boroumandfar Z, Kianpour M, Zargham A, Abdoli S, Tayeri K, Salehi M, et al. Changing beliefs and behaviors related to sexually transmitted diseases in vulnerable women: A qualitative study. Iranian journal of nursing and midwifery research. 2017;22(4):303.

24. Alam N, Streatfield PK, Khan SI, Momtaz D, Kristensen S, Vermund SH. Factors associated with partner referral among patients with sexually transmitted infections in Bangladesh. Social science & medicine. 2010;71(11):1921-6.

25. World Economic Forum, Global Gender Gap Report 2021, Insight Report. Geneva, Switzerland: World Economic Forum, 2021

26. Organization IL. Labor force participation rate, female (% of female population ages 15+) ( modeled ILO estimate)- Bangladesh January 29, 2021. Available from: https://data.worldbank.org/indicator/SL.TLF.CACT.FE.ZS?locations=BD

27. Osborn D, Cutter A, Ullah F. Universal sustainable development goals. Understanding the Transformational Challenge for Developed Countries. 2015

34. Hernán MA, Hernández-Díaz S, Werler MM, Mitchell AA. Causal knowledge as a prerequisite for confounding evaluation: an application to birth defects epidemiology. American journal of epidemiology. 2002;155(2):176-84.

35. Victora CG, Huttly SR, Fuchs SC, Olinto M. The role of conceptual frameworks in epidemiological analysis: a hierarchical approach. International journal of epidemiology. 1997;26(1):224-7.

43. Haque SE, Rahman M, Mostofa MG, Zahan MS. Reproductive health care utilisation among young mothers in Bangladesh: does autonomy matter? Women's Health Issues. 2012;22(2):e171-e80.

51. Adriel Monkam Tchokossa RN B, Timothy Golfa RN M, Omowumi Romoke Salau RN M, FWACN AAO. Perceptions and experiences of intimate partner violence among women in Ile-Ife Osun state Nigeria. International Journal of Caring Sciences. 2018;11(1):267-78

52. Payton E, Eluka N, Brown R, Dudley WN. Women's perceptions of intimate partner violence in Zambia. Violence and gender. 2019;6(4):219-26.

53. Sanawar SB, Islam MA, Majumder S, Misu F. Women’s empowerment and intimate partner violence in Bangladesh: investigating the complex relationship. Journal of biosocial science. 2019;51(2):188-202.

54.Jesmin SS. Married women’s justification of intimate partner violence in Bangladesh: Examining community norm and individual-level risk factors. Violence and victims. 2015;30(6):984-1003.

55. Silverman JG, Gupta J, Decker MR, Kapur N, Raj A. Intimate partner violence and unwanted pregnancy, miscarriage, induced abortion, and stillbirth among a national sample of Bangladeshi women. BJOG: An International Journal of Obstetrics & Gynaecology. 2007;114(10):1246-52.

Response to Reviewer Comments

1. Line 34 lacks and between cross-tab and logistic regression

Response: Thank you. We have already amended it in the Round 1 revision. We have removed the “ cross-tab, logistic regression” (line 34 in track changed version). We have now included “cross-tabulation, the conceptual framework technique and multivariable multilevel mixed-effect logistics regression” ( line 38 in track changed version; line 30 in clean version)

2. Line 37, express figures in the same format. I see that some of them presented to one decimal place yet others are presented to two decimal places. Make this consistent throughout the manuscript

Response: Thank you for highlighting this important concern. We appreciate it. We have presented numbers in a similar format. (lines 47-53 in track change version; lines 33-39 in clean version)

3. Line 57 should be women’s well-being 

Response: Thank you. We appreciate it, but we have already revised our introduction in Round 1 (lines 81-97, 107-115,124-144, 151-161,167-171 included in Round 1 ). We have already removed “women well-being” ( line 78 track changed version) and revised 1st paragraph ( lines 81-97 in track changed version; lines 49-65 in cleaned version). 

4. Line 62 : reported to lead to

Response: We are grateful for your consideration. However, our introduction was previously revised in Round 1(lines 81-97, 107-115,124-144, 151-161,167-171 included in Round 1). We've previously removed “line 62” ( line 100 track changed version) and included revised sentences (lines 107 to 115 in track changed version; lines 66-74 in cleaned version).

5. Line 105: Outcome variables

Response: Many thanks for your comment. We have amended the statement (line 196 in track changed version; line 135 in cleaned version )

6. In table 1, I think you refer to recoded not recorded

Response: We appreciate your comment. In table 1, now we have changed “recored” to “recoded”.

7. In table 3, control and decision making variables are not presented well. The names of the categories are cut – some words are below the lines. I think you can consider having the table in a smaller font so that the names of the variables and their categories appear in whole.

Response: We appreciate this concern. We have corrected it (both in track changed version & cleaned version)

8. In table 4, the word is reference not referent

Response: We appreciate this concern. But we have already updated table 4 according to the Round 1 revision. We also changed “referent” to ”reference” ( Table 4 both in track changed version & clean version). We have also format Table 3 and Table 4 ( both in track changed version & clean version ) 

9. Line 213, use have instead of has 

Response: Thank you for the suggestion. We have already updated line 213 according to the Round 1 revision (lines 375-385,422-444,447-455,473-477 included in Round 1). We have now rewritten it as “ In this study, 5.59% and 10.84% of the respondents reported genital sores and abnormal genital discharge, respectively.” ( line 375-376 in track changed version; lines 252-253 in cleaned version)

10. Line 214, in not from 

Response: Thank you for the suggestion. We have already modified line 214 according to the Round 1 revision (lines 375-385,422-444,447-455,473-477 included in Round 1). We have now rewritten it as “ Women who actively participated in joint decision-making with their husbands or partners regarding their own and child's health were significantly less likely to report STI symptoms.” ( line 378-379 in track changed version; lines 255-256 in cleaned version)

11. Line 239, delete help

Response: Thank you for the suggestion. We have already modified line 239 according to the Round 1 revision (lines 375-385,422-444,447-455,473-477 included in Round 1). We have now rewritten it as “ Receiving support from the husband benefits women in terms of better resource utilisation and the uptake of necessary health services” ( line 391-392 in track changed version; lines 267-268 in cleaned version)

12. You need a final English edit for the manuscript

Response: Thank you. We have adequately corrected the English mistake.

---

## [Decision Letter · Decision Letter 2]

17 Dec 2021

PONE-D-21-00487R2Women empowerment and sexually transmitted infections: Evidence from Bangladesh demographic and health survey 2014PLOS ONE

Dear Dr. Khan,

Thank you for submitting your manuscript to PLOS ONE. After careful consideration, we feel that it has merit but does not fully meet PLOS ONE’s publication criteria as it currently stands. Therefore, we invite you to submit a revised version of the manuscript that addresses the points raised during the review process.

We look forward to receiving your revised manuscript.

Kind regards,

Mohammad Bellal Hossain

Academic Editor

PLOS ONE

Journal Requirements:

Reviewers' comments:

Reviewer's Responses to Questions

**Comments to the Author**

1. If the authors have adequately addressed your comments raised in a previous round of review and you feel that this manuscript is now acceptable for publication, you may indicate that here to bypass the “Comments to the Author” section, enter your conflict of interest statement in the “Confidential to Editor” section, and submit your "Accept" recommendation.

Reviewer #1: All comments have been addressed

2. Is the manuscript technically sound, and do the data support the conclusions?

Reviewer #1: Yes

3. Has the statistical analysis been performed appropriately and rigorously? 

Reviewer #1: Yes

4. Have the authors made all data underlying the findings in their manuscript fully available?

Reviewer #1: Yes

5. Is the manuscript presented in an intelligible fashion and written in standard English?

Reviewer #1: Yes

6. Review Comments to the Author

Reviewer #1: Dear Authors,

Congratulations for reaching this far.

See comments below;

The manuscript needs more editing for typographical errors. These are not major but need to be checked so that it reads better. Specifically, some sentences lack joining words, do not read well and the tenses are not correct, which I believe the authors can sort. For example the first sentence in the Statistical analysis section

1. I realize that you use STI and STIs as an acronym for sexually transmitted infections interchangeably, I suggest you use one for consistency.

2. Additionally, LMIC/LMICs/LMIC countries are used interchangeably. There is need to check them for consistency

3. In the Abstract, What does “conceptual framework technique” mean

4. Line 55, begin the sentence with A instead of In, so the sentence should read “A systematic review of studies in 30 low- and middle-income countries (LMIC) showed that STI among women remain ……”

5. In table 2 change “Women participate in a variety of decision-making processes” to “Women’s participation in decision-making”

7. PLOS authors have the option to publish the peer review history of their article (what does this mean?). If published, this will include your full peer review and any attached files.

Reviewer #1: No

---

## [Author Response · Author response to Decision Letter 2]

22 Jan 2022

Comment 1 : Congratulations for reaching this far.

The manuscript needs more editing for typographical errors. These are not major but need to be checked so that it reads better. Specifically, some sentences lack joining words, do not read well and the tenses are not correct, which I believe the authors can sort. For example the first sentence in the Statistical analysis section

Response: We are grateful for your consideration. As per your feedback, we have eliminated the typographical errors and performed a thorough line-by-line edit of the manuscript to facilitate better expression. 

Comment 2 : I realize that you use STI and STIs as an acronym for sexually transmitted infections interchangeably, I suggest you use one for consistency.

Thank you for your suggestion. Now we have used the term STIs consistently in the manuscript. We have also described STI as STIs and also written symptoms of STIs in line 69, 73, 77, 78, 84, 112, 123, 126, 130, 136, 163, 180, 279, 282, 284, 349, 350, 353,355, 365,367, 372, 374,382. 

Comment3 : Additionally, LMIC/LMICs/LMIC countries are used interchangeably. There is need to check them for consistency

Response: We appreciate your comment. We have replaced LMICs as LMIC in line 69, line 97. 

3. In the Abstract, What does “conceptual framework technique” mean

Response: Many thanks. Conceptual frameworks assist the use of multivariate approaches by managing complicated hierarchical interrelationships between variables in light of social and biological knowledge. The conceptual framework depicts causal pathways between the outcome and explanatory variables based on empirical knowledge from previous study findings and the researchers’ experience. The factors included in the multivariable model have been chosen using this causal / hierarchical relationship. However, it seems that the word “technique” has added some confusion. That is why, in the manuscript, we have just included “conceptual framework” instead of “conceptual framework technique” (lines 157- 168). 

35.Victora CG, Huttly SR, Fuchs SC, Olinto M. The role of conceptual frameworks in epidemiological analysis: a hierarchical approach. International journal of epidemiology. 1997;26(1):224-7.

4. Line 55, begin the sentence with A instead of In, so the sentence should read “A systematic review of studies in 30 low- and middle-income countries (LMIC) showed that STI among women remain ……”

Response: Thank you for highlighting the issues. We changed it in Round 1, and it is currently written in the form you suggested. That is why it has remained unchanged in this version. 

5. In table 2 change “Women participate in a variety of decision-making processes” to “Women’s participation in decision-making”

Response: Thank you. We have changed “Women participate in a variety of decision-making processes” to “Women’s participation in decision-making” in Table 2, Table 3, Table 4.

---

## [Editor Report · Decision Letter 3]

2 Feb 2022

Women empowerment and sexually transmitted infections: Evidence from Bangladesh demographic and health survey 2014

PONE-D-21-00487R3

Dear Dr. Khan,

We’re pleased to inform you that your manuscript has been judged scientifically suitable for publication and will be formally accepted for publication once it meets all outstanding technical requirements.

Kind regards,

Mohammad Bellal Hossain

Academic Editor

PLOS ONE
---

## [Editor Report · Acceptance letter]

7 Feb 2022

PONE-D-21-00487R3 

Women empowerment and sexually transmitted infections: Evidence from Bangladesh demographic and health survey 2014 

Dear Dr. Khan:

I'm pleased to inform you that your manuscript has been deemed suitable for publication in PLOS ONE. Congratulations! Your manuscript is now with our production department. 

Kind regards, 

on behalf of

Dr. Mohammad Bellal Hossain 

Academic Editor

PLOS ONE